# Global-scale control of extensional tectonics on $CO_2$ earth degassing

Giancarlo Tamburello [1], Silvia Pondrelli[1], Giovanni Chiodini [1] & Dmitri Rouwet[1]

Earth degassing of $CO_2$-rich fluids has been proven to contribute significantly to the global carbon budget. The presence of ubiquitous outgassing reveals some degree of permeability of the crust that often coincides with seismically active zones. In this study, we took advantage of the most recent global geological datasets to better understand earth degassing and how it correlates with tectonic regimes. Here we use an ad hoc point pattern analysis to show that there is a spatial correlation between $CO_2$ discharges and the presence of active fault systems, in particular with those characterized by a normal slip type. Seismic data demonstrate the existence of a positive spatial correlation between gas discharges and extensional tectonic regimes and confirms that such processes would play a key role in creating pathways for the rising gases at micro- and macro-scales, increasing the rock permeability and connecting the deep crust to the earth surface.

---

[1] Istituto Nazionale di Geofisica e Vulcanologia, Sezione di Bologna, via Creti, 12, 40128 Bologna, Italy. Correspondence and requests for materials should be addressed to G.T. (email: giancarlo.tamburello@ingv.it)

Carbon dioxide is one of the most abundant natural gases in the world. It has played and still plays a crucial role in controlling temperatures on the earth surface via the greenhouse effect. During the Eocene, for instance, a postcollisional prograde regional metamorphism is thought to have released enough $CO_2$ to trigger a greenhouse global warming during this epoch[1]. Present-day natural $CO_2$ degassing is continuously updated with direct measurements and global extrapolations. The measured $CO_2$ emissions originate from volcanic or nonvolcanic sources. In the last decades the number of investigated nonvolcanic carbon-rich discharges has risen exponentially; results from these studies emphasize the important contribution to the earth gas budget.

To our knowledge, tectonic degassing has been quantified with a fairly good level of detail (albeit with some assumptions and extrapolations) for three regions in the world: the Himalayas[2] ($\sim 7.5 \times 10^5$ $km^2$ emitting $\sim 40$ Mt yr$^{-1}$ of $CO_2$), central Italy[3] ($\sim 0.4 \times 10^5$ $km^2$ emitting $\sim 10$ Mt yr$^{-1}$ of $CO_2$) and the Eastern Ethiopian rift[4] ($\sim 1.7 \times 10^5$ $km^2$ emitting $\sim 70$ Mt yr$^{-1}$ of $CO_2$, subsequently questioned and rescaled to $\sim 20$ Mt yr$^{-1}$ by Hunt et al.[5]). On the basis of a worldwide census of continental rift lengths, Brune et al.[6] have calculated a present-day $CO_2$ flux of $\sim 40$ Mt yr$^{-1}$ emitted from 20,000 km long active rifts. All these studies provided invaluable perspectives into the present and past tectonic degassing and have important implications for modeling the global carbon cycle.

Regional $CO_2$ earth degassing can be estimated by the identification of $CO_2$-rich waters. In fact, rising $CO_2$ can be sequestered for a certain amount of time (i.e. residence time) by shallow aquifers, channelized towards the atmosphere through $CO_2$-rich springs. This process allows identifying extended areas of $CO_2$ outgassing. Chiodini et al.[3,7] have demonstrated that a carbon balance based on the isotopic composition of total dissolved inorganic carbon in aquifers can be a suitable tool for discriminating $CO_2$ among biological, carbonate dissolution and deep sources. The application of this methodology to the regional aquifers in central Italy, hosted by Mesozoic carbonate-evaporite formations, allowed identifying a deep $CO_2$ flux of $\sim 10$ Mt yr$^{-1}$. The origin of this $CO_2$ discharging in central Italy is still debated: Marini and Chiodini[8] and Chiodini et al.[3] proposed a mix of mantle and a crustal-derived components; Minissale[9] favored a mantle-derived $CO_2$ origin; Italiano et al.[10] and De Paola et al.[11] suggested that mechanical faulting energy may represent an additional source able to produce $CO_2$. The further comparison between the mapped earth degassing and seismicity in central Italy[3] revealed an important correlation between gas discharges and normal faults in the Tyrrhenian hinterland. The authors argued that, in this area, deep $CO_2$-rich fluids ascend through the interconnected network of extensional fractures and normal faults and generate the earth degassing observed at the surface. On the contrary, the triggering effect of the deep degassing on the seismogenesis likely exists in the bordering Adriatic foreland, where thrusts and low-angle normal faults may create traps in which $CO_2$ accumulates and gives rise to over-pressurized reservoirs that trigger earthquakes[12] and a reduced earth degassing at the surface. The concordance between $CO_2$ discharge, seismic activity, and major faults has also been shown at a regional scale in other geodynamic contexts[4,5]. Fault systems and associated fracturing play a fundamental role in $CO_2$ propagation by creating pathways extending from the deep crust to the earth surface[5,13,14].

The first census of the main tectonic degassing areas in the world was obtained 40 years ago by Barnes et al.[15] and Irwin and Barnes[16]. Since then, the available global geological datasets and the methods of data processing have significantly improved. In this paper we update and re-examine the

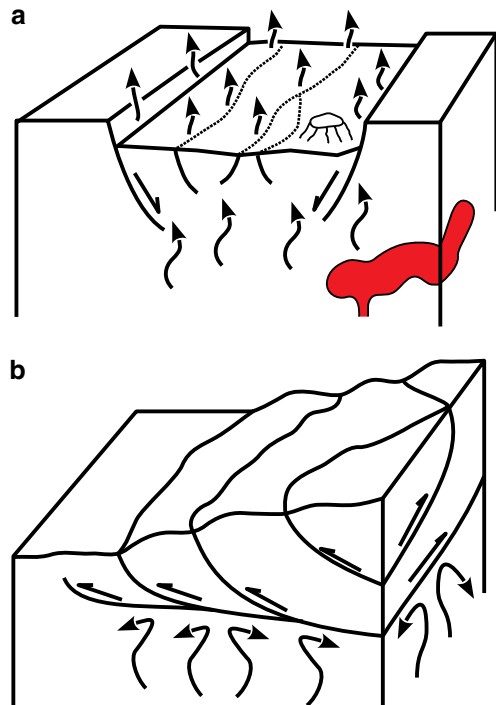

**Fig. 1** Earth degassing and tectonic regimes. **a** Sketch of an extensional tectonic setting with the pathways of fluids rising through the fractures. **b** Compressional tectonic setting with overlapping structures that may inhibit the rise of deep fluids

distribution of the main gas discharges in the world in relation to earthquakes and active faults. We observe that extensional tectonic facilitates the opening of pathways for deep rising $CO_2$-rich fluids, unlike the compressional tectonic that may hinder the transport of fluids (Fig. 1). Studying the origin of this gas is beyond the scope of this paper that does not take into account the chemical and isotopic composition of the gas.

## Results

**Carbon dioxide discharges correlated with tectonic regimes.** $CO_2$ discharges and major faults are clustered around the active continental margins (Fig. 2), with the former more concentrated in Central Eastern Europe and western United States and almost absent in cratonic areas (e.g. Australia, northern Asia). Major faults exhibit clustering along the main plate boundaries with marked hiatus between clusters which, in the majority of the cases, is also manifested in the same way by the $CO_2$ discharges (e.g. western US coast and northern Italy). There are very few gas discharges far from the mapped major faults (e.g. northeastern North America, Pannonian basin) that could be due to lack of data on fractures and faults in those areas; hence, it does not exclude the existence of a system of fractures or paleo-faults. The gas discharges dataset also shows a likely degree of incompleteness, as suggested by the scarcity of degassing sites in the African Rifts. The calculated distances (in km) between each $CO_2$ discharges and the nearest fault are <100 km for the majority of the sites (Supplementary Fig. 1). This value is of the same order of magnitude as the error in the digitization of the points suggesting a likely spatial correspondence between the two processes. Moreover, because fault systems are often characterized by a complex geometry of fractures and secondary faults that increase their lateral

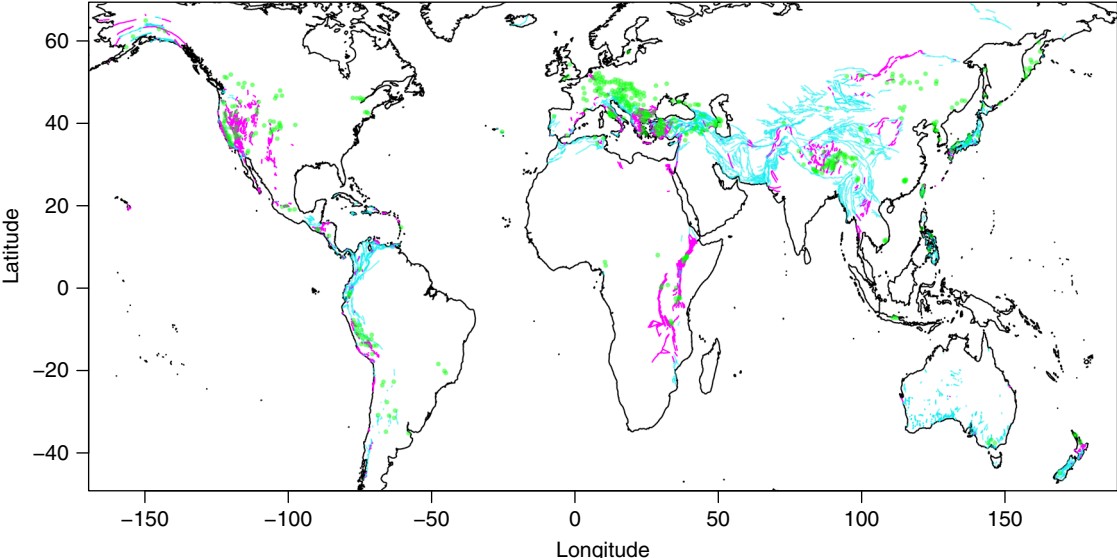

**Fig. 2** Earth degassing distribution and active faults. Distribution of carbon dioxide discharges (green dots) and active faults with (magenta lines) and without (turquoise lines) normal component

extension[17], and $CO_2$-rich aquifers may cover large areas, their proximity is plausible.

If we consider only the active faults with a normal component (magenta lines in Fig. 2), the above-described spatial correspondence with gas discharges is even more pronounced. Normal faults are spatially nearer to the gas discharges than transcurrent or thrust faults (Supplementary Fig. 1). We therefore applied the statistical test described in Methods to quantify and assess the correlation between these phenomena. We calculated a Spearman's rank correlation coefficient $\rho$ and $p$ values of gas discharges vs. all the active faults and five faults subcategories (Fig. 3a, b) divided as follows: normal faults (NF), normal faults with strike-slip component or transtensive faults (NS), strike-slip faults (SS), thrust faults with strike-slip component or transpressive faults (TS), thrust faults (TF). The NS and TS categories represent mixed-mode faulting. The subcategories have been chosen following the earthquakes tectonic regimes described by Zoback[18] and discussed below. The null hypothesis of zero correlation is predominantly rejected (>75% of the built 500 hexagonal grids) for hexagon resolutions of 4−6, the most representative of a regional degassing, but also for smaller (7) and bigger (2–4) hexagonal areas. Resolutions 8 and 9 are too small for providing statistically significant counts, given that they include only one gas discharge in the majority of the cases (>80%, Supplementary Fig. 2). These results would confirm the existence of a significant correlation between faults and gas discharges locations. In particular, normal and strike-slip faults show a better correlation compared to the thrust and transpressive faults. Thus we infer that, where an input of deep crustal or mantle-derived $CO_2$-rich gas occurs, the emplacement of the outgassing is controlled at a global scale by regional normal/transcurrent faulting and its resulting increase of crust permeability[19–22].

Our results are in line with the possible link between continental rifts lengths and their associated mantle-derived $CO_2$ degassing rates, recently proposed by Brune et al.[7]. In fact, the magnitude of the extensional faulting of the rifts would control the amount of $CO_2$ released. Furthermore, deep mantle degassing in strike slip faults has been observed at important fault systems such as the San Andreas Fault System[23] or the Norwest fault zone in Australia[24].

The tectonic control on the fluid pathway has been extensively observed in volcanic areas at regional and local scales[14,25–28], but its role at a continental scale is still poorly studied.

At the time of writing, the GEM-GAD was the only available dataset concerning faults and was focused on active faults for promoting seismic hazard modeling. Hence, we are not able to determine here the role of nonactive faults that are not currently mapped at a global scale. Nonactive faults may still operate as fluid pathways in regions where there are no (or moderate) active tectonic structures.

In light of the direct link existing between extensional faults and gas discharges, we further tested the robustness of our hypothesis by investigating more in detail the correlation with earthquakes. The instrumentally recorded seismicity allows to assess the focal mechanisms of earthquakes and hence the different tectonic regimes, or style of faulting, of recent active faults[18] (see faults classification used above). For this analysis, we used data from the Global CMT catalog[29,30] (http://www.globalcmt.org/) which includes more than 48,000 earthquakes with the relative source parameters. The moment tensors are crucial for identifying the tectonic style regime of each earthquake. As observed by Zoback[18], thrust and strike-slip regimes (TF, SS, and TS) prevail in intraplate regions (with mainly horizontal stress fields). Normal faulting regimes (NF and NS) occur mainly in topographically high areas. Both tectonic regimes and magnitudes are quite uniform at a regional scale and hence define tectonic style provinces as detailed in Zoback[18]. We applied the statistical analysis discussed in Methods to the gas discharges and the different tectonic regimes, defined on the basis of earthquake moment tensors, in order to understand if there is a correspondence between degassing and tectonic regimes (Fig. 3c, d). We have first removed noncrustal earthquakes (hypocentral depth > 70 km). The null hypothesis is consistently rejected only for the NF earthquakes, positively correlated at different hexagonal resolutions (2–8). A minor correlation is confirmed for transcurrent faults (NS and SS) for large hexagonal areas (resolution 2, red boxes in Fig. 3c, d). This result would finally confirm that extensional tectonic regimes (NF) can develop a permeable network of faults and fractures through which gases

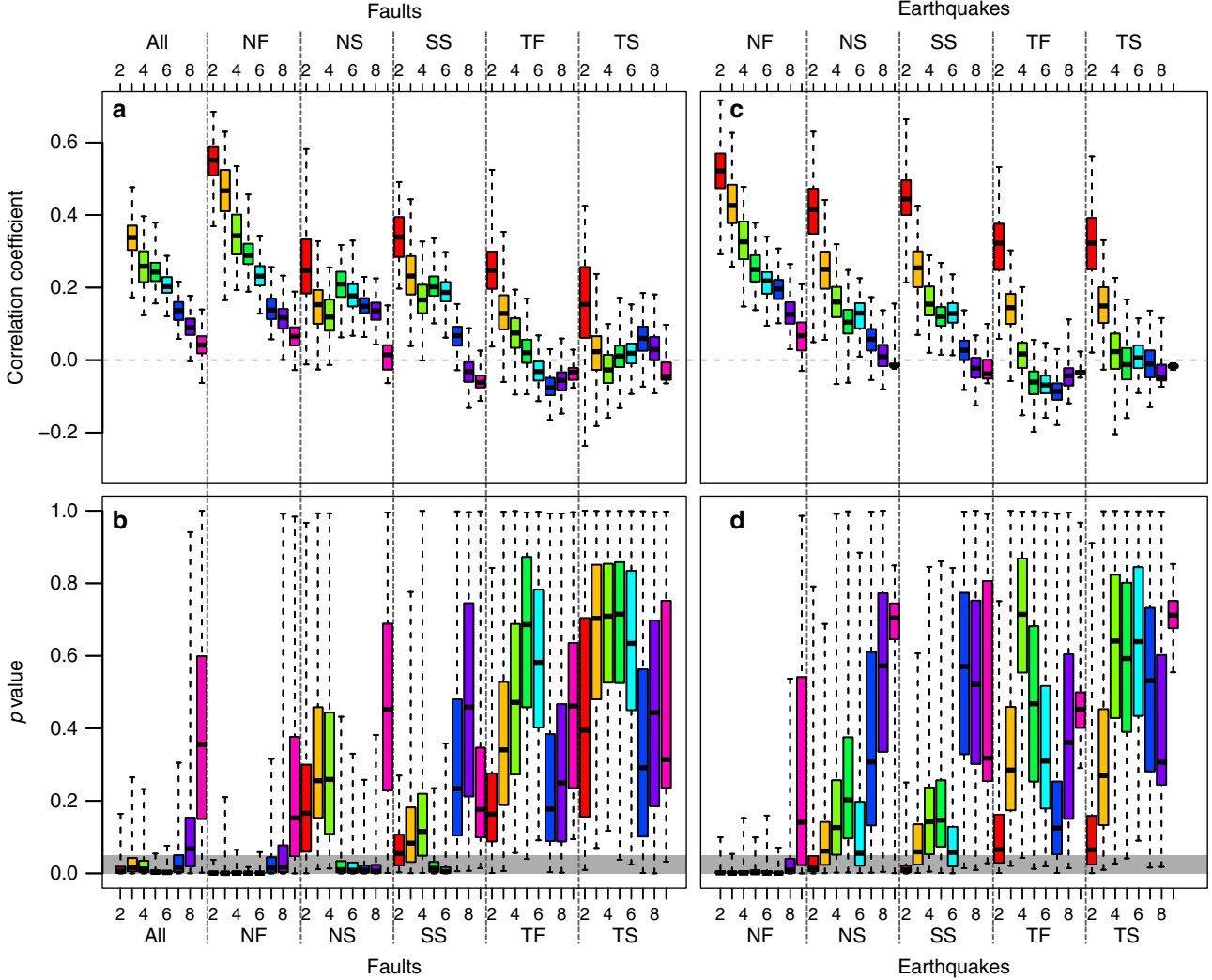

**Fig. 3** Correlation between gas discharges and tectonic regimes. Box plots of calculated p values and Spearman's rank correlation coefficients between gas discharges and **a**, **b** faults lengths of different slip types and **c**, **d** different tectonic stress regimes. Upper and lower whiskers are maximum and minimum values. Horizontal gray stripe in (**b**, **d**) shows the 95% confidence interval (the null hypothesis of no correlation is rejected if p value < 0.05). Box colors denote the eight hexagonal grids used for the calculations, from the greatest (red, resolution 2) to the smallest (pink, resolution 9) hexagonal cell

rise to the surface. Deep fault systems that penetrate the lower crust may facilitate the transfer of $CO_2$-rich gases from greater depths, upper-mantle or deep-crust[4,5] (reference therein). The encountered positive correlation between gas discharge and extensional displacements is therefore not surprising. It has been suggested that high fluid pressure at depth can play a major role in triggering earthquakes[12,22,31–34]; therefore, we argue that the positive correlation may generate also a feedback mechanism: extensional faulting facilitates the rising of deep fluids, which in turn may enhance the seismic activity.

On the contrary, thrust faulting results from compressional forces that may not significantly increase the permeability of the crust[35], here supported by the null hypothesis and negative correlation with $CO_2$ degassing features. We also stress that a separate correlation test on faults and earthquakes is mandatory. Mapped active faults may not have shown recent (M > 5) seismicity in the last decades and, vice versa, the faults that have produced recent recorded seismicity may have not been mapped yet.

A global catalog of small earthquakes (M < 5) with a high degree of completeness is not available and, hence, they are not

included in this work. But large datasets of small earthquakes at a regional scale are more easily accessible and can be representative of the local tectonic regimes[36]. As a matter of fact, Chiodini et al.[3] have demonstrated a spatial correlation between earth degassing and normal faults in the Tyrrhenian hinterland of central Italy. They argued that the triggering effect of the deep degassing on the seismogenesis likely exists in the bordering Adriatic foreland, where thrusts and low-angle normal faults may create traps in which $CO_2$ accumulates and gives rise to over-pressurized reservoirs that trigger earthquakes. The presence of these mechanisms of gas accumulation implies a reduced earth degassing in the Adriatic foreland compared to the Tyrrhenian hinterland. In light of the magnitude lower threshold of the CMT dataset and the spatial resolutions of the hexagonal grids used here, such distinction is surely not identifiable at a global scale.

**Aquifers distribution and earth degassing probabilities**. Aquifers have allowed identifying most of the tectonic degassing regions worldwide shown here[3,15,16,37], because they can trap huge amounts of rising $CO_2$-rich fluids over wide areas, bypassing the need to measure diffuse degassing at a

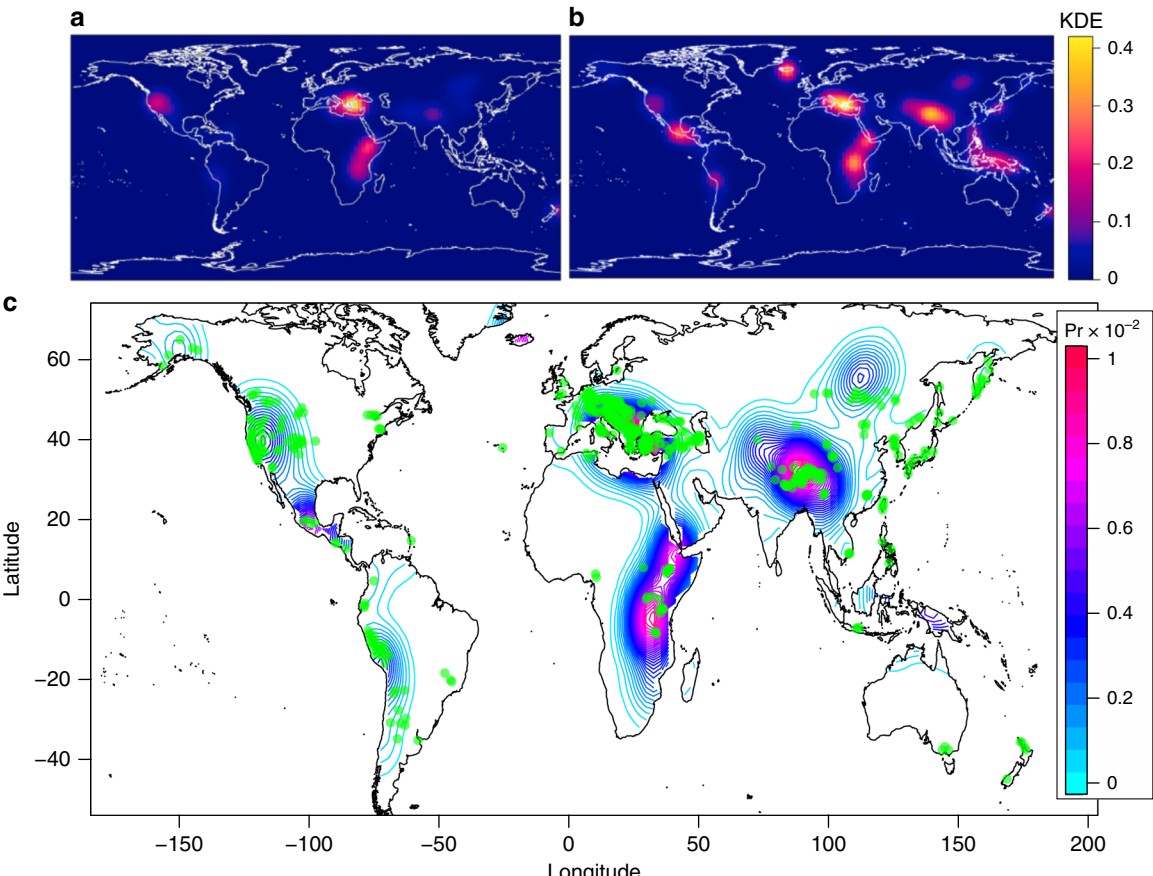

**Fig. 4** Probability of extensional tectonic and earth degassing. Estimated kernel densities for extensional faults (**a**) and earthquakes (**b**) used to obtain the probability of occurrence of tectonic gas discharges (**c**). Green dots are the existing gas discharges listed in this work

small scale in numerous sites (e.g. with an accumulation chamber[38–41]) and interpolating the results with interpolation algorithms (i.e. kriging or conditional sequential gaussian simulations[42]). The latter method is extremely time consuming in exploring large areas but it is mandatory in regions with modest or no aquifers (e.g. Ethiopian rift[4,5]). Here we took advantage of a recent estimation of the spatial distribution and volumes of modern groundwaters made by Gleeson et al.[43] in order to understand where aquifers cannot be used to study wide degassing regions. The map shown in Supplementary Fig. 3 displays the depth of the groundwater divided into five categories, as it was extracted and pooled at the land surface, and the gas discharges. The 0–0.1 m category highlights very well the arid regions that range from the western Sahara desert to the eastern Taklimakan desert and shows a marked absence of gas discharges. These arid regions may represent a threshold for our ability to observe degassing areas via carbon-rich aquifers in eastern Turkey, eastern Himalaya, surrounding Mongolia and, even if less constrained, in northern Africa and southern Australia. These evidences raise questions on the extent of earth degassing. Is the absence of gas discharges in arid regions of the world due to the absence of aquifers that help to localize them, or is it a real existing distribution of earth degassing? Below we will try to answer this question by estimating the probability of occurrence of gas discharges in these arid regions and in the rest of the globe.

We combined normal and transtensive active faults and NF earthquakes spatial distributions (those which have rejected the null hypothesis in Fig. 3) to provide a first-order probability map

of having an extensional tectonics process on Earth. In light of the positive spatial correlation between gas discharges and tectonic that we have demonstrated above, we hypothesize that higher probabilities of having extensional tectonic processes would fairly correspond to higher probabilities of having a related gas discharge. We first computed the kernel density estimates[44,45] (Fig. 4a, b) of faults and earthquakes, that is represented by a $128 \times 128$ matrix containing the expected number of normal faults and NF earthquakes per unit area. We then normalized the two matrices by dividing by their respective integrals, that are the expected global total number of normal faults and NF earthquakes, in order to obtain two probability density matrices $P(F)$ and $P(E)$ respectively. We then combined these probabilities as $P(F) + P(E) - P(F) \times P(E)$ to obtain $P(F \text{ or } E)$, that is the probability to observe normal faults or NF earthquakes in different areas of the globe (Fig. 4c). The use of the logical operator *or* means that faults and seismicity can or cannot occur in the same place and therefore are not mutually exclusive. In fact, as mentioned above, gas discharges may occur in faulted regions with no recent seismicity or in areas with recent seismicity where active faults have not been mapped yet.

The result is shown in Fig. 4c. Higher probability of having extensional tectonic processes and, hence, correlated tectonic gas discharges are found in North America, Central America, Cordillera Blanca (Perú), central Europe, East African Rift and eastern Himalaya, and correspond well to the higher density of reported gas discharges (green dots). With the exception of Melanesia, there are no wide areas with noticeable high probabilities and scarcity of mapped gas discharges,

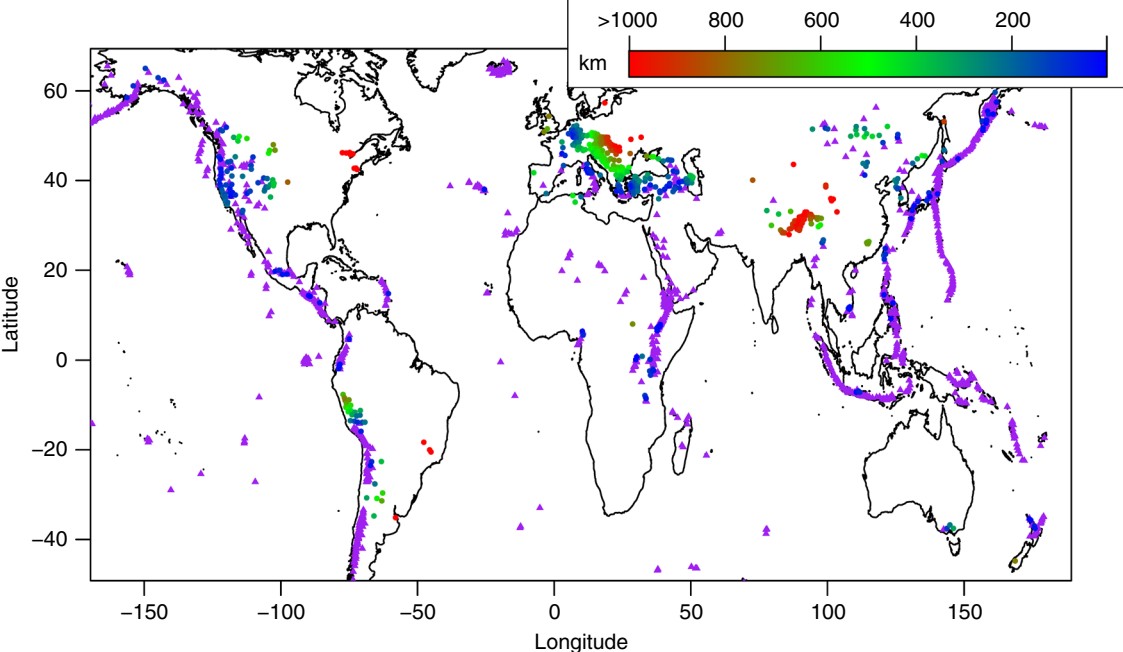

**Fig. 5** Volcanic influence on earth degassing. World map of Holocene volcanoes (purple triangles) and gas discharged (circle) colored in function of the distance from the closest holocene volcano

suggesting a certain degree of completeness of the gas dataset here re-processed or, if not complete, at least representative of the current global tectonic degassing. Concerning the arid region discussed above, there is no evidence of a high probability of gas discharges occurrence. We therefore argue that there may be no relevant tectonic degassing that has not been studied yet. Instead, further studies on the tectonic degassing along the Ethiopian rift, central America, southern Central Asia (e.g. Kyrgyzstan and Tajikistan), western East Asia (i.e. eastern Himalaya) and Melanesia (poorly constrained) are strongly recommended based on the existing contrast between high probabilities and low density of mapped degassing sources. It should be underlined that the intensity of the degassing process is not determined for each gas discharge reported here, and one single point could be representative of an intense degassing process operating over a wide region. But we believe that the extension of the degassing area or aquifer to which each point belongs cannot be clearly observable at a global scale and that is adequately represented by the dimension of the symbols in the figures. Accordingly, the gas discharges density can be considered a good indicator of the intensity of degassing process or of the lack of data in some regions.

For completeness, the same procedure has been also applied for compressional tectonic regimes. The result (Supplementary Fig. 4) shows lower spatial correspondence between the main degassing regions and the areas with high probability of compressional tectonic, further corroborating our findings above discussed.

**Influence of volcanism on tectonic degassing.** In the discussion above, not much has been said about the origin of the gas. In fact, as already mentioned, any attempt to identify the source of the discharged gas would primarily require the knowledge of its isotopic and chemical composition[4,7,21]. Numerous studies have already investigated the origin of the gas in several regions and creating a database with such measurements, despite

challenging, would be possible. But for a quick investigation on the magmatic origin of the gas discharges here reported, we can tentatively use a list of Holocene active volcanoes[46] considering that a volcanic edifice can be an indicator of magmatic processes occurring nearby and leading to $CO_2$ degassing[5]. The dataset includes 1444 volcanoes formed during Holocene, among which ~400 have erupted in the last century. Therefore, for each emission point we have calculated the minimum distance (in kilometers) from a volcano. The resulting map in Fig. 5 clearly shows that three tectonic degassing regions in the world (roughly corresponding to the Cordillera Blanca in Perú, Pannonian basin in Central Eastern Europe and eastern Himalaya) are separated by long distances (>400 km) from Holocene volcanoes suggesting a nonmagmatic origin of their gas discharges. As a matter of fact, the origin of the gas released from these three regions has been widely documented in previous works as described below.

The Cordillera Blanca is a massif in the Peruvian flat slab segment (a distinct type of low-angle subduction[47]) characterized by an absence of active continental arc magmatism and a millions years old undertaking extensional regime[48]. The helium isotopic data from the thermal springs issuing from this region provided direct evidence for circulation of mantle-derived fluids in an amagmatic flat-slab setting[49]. The more southern Pampean flat-slab segment of the Central Andes displays a similar scenario of the Cordillera Blanca, but with minor current extensional tectonics[50] and known degassing activity[15].

The Himalayas are characterized by diffuse gas emissions[51] and hot springs[52,53] with high $\delta^{13}C$ values of the $CO_2$ gas phase suggesting a deep metamorphic decarbonation and a subsequent release of $CO_2$-rich gas[2,54]. An increase of diffuse $CO_2$ emissions has been observed after the 2015 earthquake in Nepal[55].

The Pannonian Basin is a Miocene back-arc basin characterized by several past extensional events and a high geothermal gradient[56] (references therein). The numerous gas reservoirs scattered over the basin have been intensively studied[57–59] and

provided convincing evidence for a significant component of mantle-derived carbon[21].

All of the above features are indicative for a nonmagmatic origin of these three degassing regions that have been recognized here only on the basis of the distance from Holocene volcanoes.

In this work, we have complemented and re-analyzed the census of carbon dioxide discharges of Barnes et al.[15] and Irwin and Barnes[16] and provided new information on tectonic degassing and its mechanisms and the factors that control its distribution on Earth. Although a correlation between tectonic degassing and earthquakes has been already proposed and qualitatively observed, we have provided further evidence for such correlation and described more in detail the crucial role of the extensional regimes in facilitating gas migration. We found out that more efforts should be dedicated to better constrain the tectonic degassing in central America, southern Central and western East Asia. Finally, we argue that future estimations would not change significantly the magnitude of the global degassing that has been hypothesized in the last decade (hundreds of megatons per year of $CO_2$).

## Methods

**World datasets used**. A large part of the gas data reported in this work was available in papers only in the form of plots and maps with no numerical table as reference. Hence, we have used an open source software[60] for automatically digitizing the images and extracting geographic coordinates of points. We estimated the error of the imported coordinates by digitizing known coordinates from a world map. The error of ~0.1° longitude and ~0.4° latitude would correspond to distances ranging from 44.8 to 45.4 km, depending on the latitude. We consider such error not relevant for our analysis at a global scale, but we do not recommend the use of such data for studies at smaller scales. Most of the carbon dioxide discharges have been digitized from Barnes et al.[15] and Irwin and Barnes[16] and correspond to water springs with dissolved $CO_2$ or $HCO_3^-$ > 1 g l$^{-1}$ and with a pH < 8.3. We integrated the complementary gas discharges in Italy available in Googas[61] and MAGA[62] databases that are defined as nonvolcanic and high flux emissions. Other gas discharges have been imported for Portugal[63], Tibet[53] and Cameroon[64]. All the gas discharges coordinates used in this work and their references are summarized in Supplementary Data 1.

The world active faults dataset was obtained from the Global Earthquake Model Foundation's Global Active Faults project (GEM-GAF), freely downloadable at https://github.com/cossatot/gem-global-active-faults. The dataset consists of GIS files containing fault traces and small amounts of relevant attributes or metadata (fault geometry, kinematics, slip rate, etc.). However, given the absence of data in Melanesia and southern East Asia, we digitized and added the main fault systems described in Abers and McCaffrey[65] and Darman and Sidi[66] respectively. Gas discharges (green circles) and active faults (magenta and turquoise lines) are shown in Fig. 2.

Seismic data were obtained from the Global CMT Catalog (http://www.globalcmt.org/), a collection of seismic moment tensors of great to moderate magnitude earthquakes ($M \geq 5.5$) occurred worldwide since 1976 to present[29,30].

A list of 1443 volcanoes with eruptions during the Holocene period (~11,700 years ago) has been obtained from the database of the Global Volcanism Program[46].

**Spatial analysis of world datasets**. In order to assess the correlation among these different spatial datasets, we applied a point pattern analysis. In this context, a common type of analysis that can be performed is quadrat counting. Quadrat analysis consists of dividing a study area into equal sized subareas called quadrats and using probability analysis to determine and correlate the actual frequency of points within each quadrat. However, dividing a world map into large quadrats of equal longitude and latitude intervals would generate areas of unequal dimensions and geometries. Therefore, a grid that can divide the whole globe into areas of similar extension is needed. Researchers in numerous fields[67] have consistently concluded that hexagons are the optimal choice for discrete gridding at a global scale. To achieve this, we used a package for the R programming language called "dggridR"[68,69] that allows dividing the entire globe into hexagons of equivalent extension, and posteriorly count the data points in each area for performing ulterior statistical analysis. The extension of the hexagons can been chosen among 21 levels (called "resolution", see Supplementary Table 1), ranging from millions of square kilometers (level 0) to hundreds of square meters (level 20). As for quadrat analysis, the selection of the extension of the quadrats/hexagons can significantly influence the outcome of a statistical analysis. For our analysis we selected a resolution ranging from 2 to 9 corresponding to areas of ~5,700,000 to ~2600 km$^2$, and average radius of ~1300 to ~28 km (Supplementary Table 1). However, we consider the hexagons with resolutions 4−6 (~630,000 to ~70,000 km$^2$, and average radius of

~450 to ~150 km) of the same order of magnitude of the extensions of the different degassing regions and faults systems in the world (e.g. Chiodini et al.[3] for central Italy) and hence the most representative for such analysis. We have then discarded the hexagons without gas discharges. A typical example of global hexagonal grids is shown in Supplementary Fig. 5. The grid can be shifted by changing longitude and latitude of the starting point from which it is built (by default 58.3° and 11.25°). Hence, we have randomly changed these values from −10° to 10° around the default starting point in order to obtain 500 grids; in fact after a shift of 20° the grid simply repeats. Such randomization is proposed also for quadrat subdivision[70]. For each grid we have counted the number of gas discharges, earthquakes (of different magnitudes and tectonic regimes) and lengths of major faults within each hexagon. This procedure allows avoiding the possible bias due to the selection of one single position of the grid. As example, in Supplementary Fig. 6 we show a scatterplot between gas discharges counts and earthquakes counts obtained by counting the points within the hexagons of a grid of resolution 2 (among 500 randomized grids). Given the non-normality of the distributions of count data (gas discharges and earthquakes have a negative binomial distribution, see Supplementary Fig. 2) and the presence of continuous variables (lengths of faults), a Spearman's Rank correlation coefficient $\rho$ has been preferred for calculating the correlation between gas discharges counts and the counts of earthquakes and lengths of active faults for each of the 500 different grids generated for each resolution in the 2−9 interval. Then we tested a null hypothesis that states that each value of $\rho$ is significantly (95% confidence interval) equal to zero, and hence, no correlation exists. We calculated a $p$ value for each correlation test and set a predefined threshold value to 0.05 for rejecting the null hypothesis. If the null hypothesis is rejected ($p$ value < 0.05), the alternative hypothesis is, instead, accepted and states that $\rho$ is significantly different than zero and hence a correlation may exist. Calculated $\rho$ and $p$ values are shown in the box and whiskers diagram in Fig. 3.

## Data availability

The authors declare that the data supporting the findings of this study are available within the supplementary information files.

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

## Acknowledgements

This work has been supported by the Deep Carbon Observatory (subcontract N. 10759-1254 "A first step toward the estimation of global tectonic carbon flux", Principal

Investigator: Carlo Cardellini). The authors wish to thank Tobias Fischer and an anonymous reviewer for their useful comments and suggestions upon reviewing the manuscript. Antonio Costa and Carlo Cardellini are also thanked for their valuable discussions that led to conceiving this manuscript.

## Author contributions

G.T. conceived the idea for this manuscript and extracted the datasets and performed the statistical analysis. S.P. suggested to focus on the tectonic regimes and provided and processed the seismic data from GMT catalog. G.T. wrote the manuscript and S.P., G.C. and D.R. participated in the discussion and revised the text.

## Additional information

**Competing interests:** The authors declare no competing interests.

