## [Peer Review File · Nature Communications]

Reviewers' comments:

Reviewer #1 (Remarks to the Author):

This is a very interesting paper that utilizes a statistical discretization of the planet to determine if or how CO₂ degassing correlates with active faulting and volcanoes active in the Holocene. In some respects, this is a much-needed update of the original seminal work of Barnes and Irwin in the 1980's, but with better constraints on the correlation between active faulting and earth degassing. The results of this study show that normal faulting (extensional) environments show the strongest correlation, followed by strike-slip and then thrust faulting. That the authors find correlations is important because that probably has direct implications for an influence of deeply buried CO₂ and earthquakes.

In my opinion, this paper can be published, and will find a wide-ranging and interested audience. The authors also predict that future studies will most likely verify that the natural global CO₂ degassing will converge to a few hundred megatons/year. This is minute compared to anthropogenic CO₂ contributions to the atmosphere, but is nevertheless an important value to constrain.

other comments:

1) The paper is well-written for the most part, but I have a pet peeve with sentences that include, for example, "this procedure allowed to avoid...", when it sounds better as "this procedure allows avoiding...". There are many such examples in the text, and I recommend these be changed (e.g. lines 47, 54, 130, 224, 278

There is no verb in the sentence on line 60.

line 125, explain more what you mean by extreme distortion.

Figures are not introduced in order.

line 135. Change extensively to significantly

line 218. Change 'is' to 'are'

line 259. Change to February 22, 2011.

line 263. datasets

line 280. amounts

line 301. tectonics

line 352. Holocene

line 390. Right parentheses is missing.

Reviewer #2 (Remarks to the Author):

by Tobias Fischer

This paper provides and new and innovative analyses of the global correlation of deep earth CO₂ degassing with faults. While this correlation has been pointed out by several authors and in

different tectonic settings (arcs, rifts) the global evaluation of this correlation is new and very significant for the geoscience audience. The paper should be published in nature communications. However, the following issues need to be adequately addressed:

1. the main point of the paper - the correlation of deep CO₂ degassing with extensional faulting needs to be more clearly stated. To this end Figure 1 is not very helpful and does not contribute to the clarity of the paper. This figure could be replaced with a schematic figure of the tectonic regimes that shows CO₂ degassing from subduction zones, convergent zones and extensional rifts. That figure could then be used to state the goal of the paper, i.e. to correlate CO₂ degassing with extensional faulting. Reference could also be made here to the recent Burne et al., 2017 Nat Geosci paper that aims to evaluate rifting CO₂ contribution to global climate fluctuations in the past.

2. related to point 1 above, the methodology of the statistics of the correlation (lines 152 to 154, specifically) need to be more clearly explained.

3. the uncertainty of faults that are not yet mapped or faults that have not produced recent seismicity (lines 257-258) should be better evaluated. The authors point to hidden faults that produced large earthquakes and that are not in the record. If such faults cannot be quantified, then the question needs to be assessed whether this paragraph should be included because it really does not add to the argument. I would suggest to focus the argument on the faults that can be detected and the degassing features that are mapped and then highlight the correlations between the two.

4. Section 4 is very confusing and the title of this section needs to be changed to reflect the content of this section which really deals with the aquifers and the probability that high degassing regions have not yet been measured.

5. the section on final tectonic degassing budget is not helpful and simply provides a review of the literature and states that the proposed literature values are probably a good approximation however there is no rationale given from the presented work why this approximation should be valid. This section could be removed and the literature fluxes could be presented in the introduction.

Specific minor points

I. 61 ... East African Rift system received significant attention

I. 62/63 clarify that Hunt et al., measured fluxes from the Main Ethiopian Rift and that Lee et al measured CO₂ fluxes from the Natron-Magadi basins in the Eastern Branch of the East African Rift. Only Lee et al., measured isotopes. Both studies used their measurements to extrapolate to the entire East African Rift.

I. 74 ... significant insights with regards to the correlation ...

I. 80 also cite Muirhead et al., 2016 Geosphere who specifically looked at faulting and the role of fluids for extensional regions

I. 122 ... consists of dividing...

I. 214 include Muirhead et al., 2016 Geosphere paper who demonstrated that tectonic control is critical for fluid movement in extensional rifted regions.

I. 280 ... huge amounts of...

I. 320 and others. Be consistent with the region names. Peru should be Cordillera Blanca, It is not the Ethiopian Rift but the East African Rift or the Main Ethiopian Rift if it refers to the rift section south of the Afar region but north of Kenya. I think authors refer to East African Rift where they state "ethiopian rift"

POINT-BY-POINT RESPONSE TO THE REFEREES' COMMENTS OF

Global-scale control of extensional tectonics on CO₂ earth degassing

Giancarlo Tamburello¹, Silvia Pondrelli¹, Giovanni Chiodini¹, Dmitri Rouwet¹,

¹ *Istituto Nazionale di Geofisica e Vulcanologia, Sezione di Bologna (Italy)*

Reviewer #1:

1) The paper is well-written for the most part, but I have a pet peeve with sentences that include, for example, "this procedure allowed to avoid...", when it sounds better as "this procedure allows avoiding...". There are many such examples in the text, and I recommend these be changed (e.g. lines 47, 54, 130, 224, 278

We thank the reviewer for his positive appreciation of our work. We have changed “allowed to” in “allows avoiding” in the whole manuscript.

There is no verb in the sentence on line 60.

We've corrected this typo.

line 125, explain more what you mean by extreme distortion.

We've replaced “extreme distortion” with “areas of unequal dimensions and geometries”.

Figures are not introduced in order.

We added a reference to Figure 2 in section 2.

line 135. Change extensively to significantly

Ok, changed.

line 218. Change 'is' to 'are'

Ok, changed.

line 259. Change to February 22, 2011.

We decided to keep the format “day month year” for the whole manuscript

line 263. datasets

Ok

line 280. amounts

Ok

line 301. tectonics

Ok

line 352. Holocene

Ok, all “Holocene” words in the manuscript have been corrected

line 390. Right parentheses is missing.

Ok, added

Reviewer #2:

1. the main point of the paper - the correlation of deep CO₂ degassing with extensional faulting needs to be more clearly stated. To this end Figure 1 is not very helpful and does not contribute to the clarity of the paper. This figure could be replaced with a schematic figure of the tectonic regimes that shows CO₂ degassing from subduction zones, convergent zones and extensional rifts. That figure could then be used to state the goal of the paper, i.e. to correlate CO₂ degassing with extensional faulting. Reference could also be made here to the recent Burne et al., 2017 Nat Geosci paper that aims to evaluate rifting CO₂ contribution to global climate fluctuations in the past.

We thank the reviewer for his comments. We changed part of the introduction and Figure 1 in order to better emphasize the role of tectonic regimes in controlling earth degassing processes and, hence, the goal of the paper. Brune et al. (2017) was already cited in the manuscript but it was not listed in the references. Now it is added.

2. related to point 1 above, the methodology of the statistics of the correlation (lines 152 to 154, specifically) need to be more clearly explained.

We have added more information in the manuscript and another figure in the supplementary material for explaining better how we calculate the correlation coefficient and p-value.

3. the uncertainty of faults that are not yet mapped or faults that have not produced recent seismicity (lines 257-258) should be better evaluated. The authors point to hidden faults that produced large earthquakes and that are not in the record. If such faults cannot be quantified, then the question needs to be assessed whether this paragraph should be included because it really does not add to the argument. I would suggest to focus the argument on the faults that can be detected and the degassing features that are mapped and then highlight the correlations between the two.

We preferred to remove this sentence.

4. Section 4 is very confusing and the title of this section needs to be changed to reflect the content of this section which really deals with the aquifers and the probability that high degassing regions have not yet been measured.

We changed the title of section 4 and tried to link the discussions on aquifers and probability of earth degassing.

5. the section on final tectonic degassing budget is not helpful and simply provides a review of the literature and states that the proposed literature values are probably a good approximation however there is no rationale given from the presented work why this approximation should be valid. This section could be removed and the literature fluxes could be presented in the introduction.

We agree with the reviewer. We moved the literature fluxes in the introduction.

Specific minor points

1. 61 ... East African Rift system received significant attention

Ok

1. 62/63 clarify that Hunt et al., measured fluxes from the Main Ethiopian Rift and that Lee et al measured CO₂ fluxes from the Natron-Magadi basins in the Eastern Branch of the East African Rift. Only Lee et al., measured isotopes. Both studies used their measurements to extrapolate to the entire East African Rift.

Ok

1. 74 ... significant insights with regards to the correlation ...

Ok

l. 80 also cite Muirhead et al., 2016 Geosphere who specifically looked at faulting and the role of fluids for extensional regions

Ok

l. 122 ... consists of dividing...

Ok

l. 214 include Muirhead et al., 2016 Geosphere paper who demonstrated that tectonic control is critical for fluid movement in extensional rifted regions.

Ok

l. 280 ... huge amounts of...

Ok

l. 320 and others. Be consistent with the region names. Peru should be Cordillera Blanca, It is not the Ethiopian Rift but the East African Rift or the Main Ethiopian Rift if it refers to the rift section south of the Afar region but north of Kenya. I think authors refer to East African Rift where they state "ethiopian rift"

Ok